# Introduction of Infection Prevention Tracheal Intubation Protocol during the COVID-19 Pandemic Is Not Associated with First-Pass Success Rates of Endotracheal Intubation in the Emergency Department: A Before-and-After Comparative Study

**DOI:** 10.3390/jpm13061017

**Published:** 2023-06-19

**Authors:** Wooseok Jang, Hyunggoo Kang, Hyungoo Shin, Changsun Kim, Heekyung Lee, Hyukjoong Choi

**Affiliations:** 1Department of Emergency Medicine, College of Medicine, Hanyang University, Seoul 04763, Republic of Korea; 2Department of Emergency Medicine, Hanyang University Guri Hospital, Guri 11923, Republic of Korea

**Keywords:** COVID-19, emergency department, endotracheal intubation, first-pass success, intubator, infection

## Abstract

Aerosols and droplets have put healthcare workers performing airway management at high risk of contracting coronavirus disease 2019 (COVID-19). Experts have developed endotracheal intubation (ETI) guidelines and protocols to protect intubators from infection. We aimed to determine whether changes in the emergency department (ED) intubation protocol to prevent COVID-19 infection were associated with first-pass success (FPS) rates in ETI. We used data from the airway management registries in two academic EDs. The study was divided into pre-pandemic (January 2018 to January 2020) and pandemic (February 2020 to February 2022) periods. We selected 2476 intubation cases, including 1151 and 1325 cases recorded before and during the pandemic, respectively. During the pandemic, the FPS rate was 92.2%, which did not change significantly, and major complications increased slightly but not significantly compared with the pre-pandemic period. The OR for the FPS of applying infection prevention intubation protocols was 0.72 (*p* = 0.069) in a subgroup analysis, junior emergency physicians (PGY1 residents) had an FPS of less than 80% regardless of pandemic protocol implementation. The FPS rate of senior emergency physicians in physiologically difficult airways decreased significantly during the pandemic (98.0% to 88.5%). In conclusion, the FPS rate and complications for adult ETI performed by emergency physicians using COVID-19 infection prevention intubation protocols were similar to pre-pandemic conditions.

## 1. Introduction

The World Health Organization declared the novel coronavirus disease 2019 (COVID-19) a pandemic on 11 March 2020; it continues to cause numerous deaths worldwide [1]. COVID-19 can cause severe respiratory complications, and some patients require endotracheal intubation and ventilation to maintain adequate oxygen levels. COVID-19 is transmitted by aerosols and droplets, increasing the risk of infection for healthcare workers involved in airway management [2,3,4,5,6,7,8]. For this reason, some new protocols for safe endotracheal intubation (ETI) have been proposed by experts during the COVID-19 pandemic [9,10,11,12,13,14,15]. These protocols generally recommend wearing appropriate personal protective equipment (PPE), intubation by the most experienced medical personnel, limiting the number of intubation team members, using a video laryngoscope, and minimizing the production and spread of aerosols [16,17].

However, there are some potential risks associated with the use of these protocols that should be considered. Firstly, the use of unfamiliar protocols may adversely affect the behavior of the intubation team, particularly in emergency departments (EDs), where patient information is lacking and difficult situations are more likely to occur [13,18]. In addition, the limited number of intubation team members and communication difficulties due to PPE may be limiting factors [19]. In addition, protocols encourage experienced physicians to be the first intubator to attempt, which may reduce the opportunity for less experienced physicians to gain experience [20,21].

We were concerned that these potential risks might lead to lower FPS of ETIs in the ED. Multiple intubation attempts are associated with a higher risk of adverse outcomes [22,23,24]. Therefore, investigating the association between new protocols and FPS of intubation will help us prepare for other infectious disease pandemics in the future. To date, a few ED-based studies have reported their experience with ETI protocols to prevent infection during the COVID-19 pandemic [25,26,27]. However, more research is needed on this topic to draw universal conclusions that can be applied to different ED settings [4,8,25,26,27,28,29,30].

The aim of this study was to answer the following question: Is adoption of a COVID-19 infection prevention intubation protocol in the ED associated with FPS and other intubation-related outcomes?

## 2. Materials and Methods

### 2.1. Study Design

We conducted a before-and-after retrospective observational study using airway registry data from two academic EDs. These two EDs have been collecting data on airway management in the ED through the airway registry since 2011. In this study, we used data from January 2018 to January 2020 (the pre-pandemic period) and from February 2020 to February 2022 (the pandemic period). Intubation data after this period were not included because of significant changes in infection control guidelines in South Korea. ED A is a local emergency medical center, and ED B is a regional emergency medical center. Both EDs ran a 4-year emergency medicine (EM) residency training program. Each ED receives an average of 42,000 to 45,000 patients per year. In addition, almost all ETIs are performed by emergency physicians, with an average of 250 ETIs per year in each center. Typically, a senior physician supervises a junior resident during intubation.

Prior to the pandemic period, all EM residents rotated between the two EDs and attended three airway management training courses (basic, intermediate, and advanced) before starting live intubation. The courses consisted of didactics, skills training, and simulation. The participants were given ample opportunity to perform intubations on a variety of high-fidelity simulators and on cadavers using a wide range of equipment. In contrast, during the pandemic period, new EM residents attended a single, streamlined workshop consisting of lectures and skills training prior to live intubation. Lectures were delivered online, and the workshop session focused on a few key skills, including video laryngoscopy techniques, with an instructor/trainee ratio close to 1:1 to increase the effectiveness of the training.

In this study, emergency ETIs were performed according to the guidelines of the Korean Emergency Airway Management Society (KEAMS), which are broadly consistent with the recommendations of several experts published early in the pandemic [17,31,32]. In the pre-pandemic period, medical personnel involved in intubation wore only surgical masks and gloves. South Korea reported its first COVID-19 case on 20 January 2020, and ETIs were performed at our center in February 2020 in a negative-pressure isolation room with extensive PPE. PPE was equivalent to Level C, which included a surgical gown, disposable plastic apron, dual gloving, an N95 mask, and a face shield. These rules were uniformly applied regardless of the patient’s risk of COVID-19 infection. This is because any patient undergoing a procedure that could produce aerosols during the pandemic was considered a potential patient with COVID-19 [3,4,5,6,7]. We also expected that using a single protocol for all patients would help the ED staff adapt. The intubators wore powered air-purifying respirators, specifically 3M Jupiter Powered Air Turbo with a breathing tube (BT-20 L) and a loose-fitting hood (S-433 L-5) (3M, St. Paul, MN, USA), instead of face shields.

To minimize the number of ED staff exposed to aerosols, the number of people directly involved in intubation was limited to less than three. If the junior emergency physician attempted to intubate, the senior emergency physician instructed the junior emergency physician on the intubation strategy prior to intubation and then waited outside the isolation room donning PPE. We did not use an aerosol box because we predicted a significant increase in intubation difficulty for a small decrease in infection risk.

Prior to the implementation of the new protocol, mock intubation drills were conducted for emergency department staff using human patient simulators in a negative-pressure isolation room. Through these mock drills, we determined the correct equipment, staffing, and communication for our ED environment.

### 2.2. Study Population

We included patients aged ≥18 years who underwent oral ETI at either ED during the study. The exclusion criteria were as follows: (1) ETI not attempted by an emergency physician; (2) those in which surgical methods were used on the first intubation attempt; (3) those performed for tube exchange using a tube exchanger; and (4) a nasotracheal approach (Figure 1).

### 2.3. Data Collection

Each intubator completed an airway registry form developed in 2006 based on the consensus of the Korean Emergency Airway Management Society (KEAMS) investigators [25]. Both EDs have been participating in this registry since 2006 and in the second version since 2011. The ethics committees of both hospitals approved this prospective airway registry and retrospective study. After the ETI, the intubator recorded the information on the form, which was reviewed by the investigator in each ED and entered into a web-based database. Authors H.K. and H.J.C. monitored the completeness of data collection during the study period. After the completion of the data collection, the original data forms were reviewed by author H.J.C.

### 2.4. Measures

We examined the sex, age, vital signs, and oxygen saturation (SpO_2_) during the decision to intubate; indications for intubation; the predicted difficulty of glottis exposure; physiologic difficulty; crash airway; intubators’ grade; ETI method; device used in the first attempt; first-pass success (FPS) or failure; the number of ETI attempts; glottis exposure grade; and postintubation complications. One attempt was defined as a single insertion of a laryngoscope blade past the tooth. The predictors of difficult glottic exposure were defined as associated external trauma, limited mouth opening/neck movement, short hyomental/thyrohyoid distances, and symptoms or signs of upper airway obstruction. Crash airway was defined as a cardiac arrest or near-arrest situation that did not require medications or immediate intubation. We used the Cormack–Lehane (CL) classification to determine the degree of glottic exposure. Physiological difficulty was defined as a mean arterial pressure of less than 65 mmHg or an SpO_2_ of less than 90% prior to intubation. Among post-intubation complications, we defined post-intubation hypotension as mean arterial pressure <65 mmHg within 30 min of the first intubation attempt and post-intubation hypoxemia as a decrease in SpO_2_ to <90% within 30 min of the attempt.

The primary outcome of the study was the FPS rate, and the secondary outcome was postintubation complications. We also looked at changes in other ETI-related variables.

### 2.5. Statistical Analysis

We used SPSS for Windows version 28.0 (SPSS Inc., Chicago, IL, USA) for all statistical analyses. The ETI cases were divided into pandemic period (ETI during the intervention period) and control period (ETI during the preintervention period) groups. The continuous variables are presented as means and standard deviations, whereas other descriptive data are presented as frequencies and percentages. To determine the independent effect of the use of COVID-19 intubation protocols on the FPS rate, we selected the following potential factors that could influence the FPS rate by chi-squared test ED, crash airway, anatomical/physiological difficulty, glottic exposure grade, intubator grade, and intubation method/device. To determine the independent association of the COVID-19 intubation protocol and other variables on the FPS rate, we performed a stratified analysis among the independent variables and explored effect modification factors based on the homogeneity of the odds ratio. We then performed a multivariate logistic regression analysis to identify the variables. We performed stratified analysis among independent variables using univariate analysis and searched for effect-modifying factors based on the homogeneity of the odds ratio (OR). We identified all variables with *p* < 0.10 as candidate parameters for inclusion in the subsequent multivariate model. We used multivariate logistic regression to find variables that had an independent association with the FPS. Odds ratios are reported with 95% confidence intervals (CIs) and *p*-values. All reported CIs are 2-sided 95% intervals, and tests were performed at the 2-sided 5% significance level.

## 3. Results

Finally, we selected 2476 intubation cases, including 1151 and 1325 cases before and during the pandemic, respectively. Of these patients, 63.2% (1566/2476) were male, and the median age was 68 years (IQR: 56–78). We observed no differences in the age or sex of patients between the pre-pandemic and pandemic periods (Table 1). However, several intubation-related characteristics differed between the periods (Table 1). The proportion of cases caused by injury decreased significantly from 19.9% in the pre-pandemic period to 15.1% in the pandemic period (*p* = 0.002). The purpose of intubation during the pandemic did not differ from that before the pandemic (*p* = 0.885). In addition, the rate of crash airway was similar during and before the pandemic (*p* = 0.647). During the pandemic, the percentage of patients with physiological difficulties increased significantly from 19.2% to 22.6% (*p* = 0.040), while the predicted anatomical difficulties remained unchanged. The number of intubations per ED differed before and during the pandemic, with ED B reporting a significant increase from 398 to 753 cases during the pandemic (*p* < 0.0001).

In terms of intubator grade, the proportion of postgraduate year (PGY) 1 residents was significantly lower during the pandemic (22.3%) than before the pandemic (27.2%) (*p* = 0.009). In addition, the percentage of emergency physicians with a PGY 3 or higher increased from 48.1% to 53.5%. (Table 2). Among intubation methods, the proportion of awake techniques decreased from 6.9% to 4.5%, while the proportion of RSI increased from 50.0% to 52.3% (*p* = 0.037). Among the devices used for intubation, the proportion of direct laryngoscopes (DL) decreased from 9.6% before the pandemic to 6.4% during the pandemic, and the proportion of portable video laryngoscopes (VL) with disposable blades increased significantly from 11.1% to 42.7% during the pandemic (*p* < 0.0001). The usage rate of mounted VLs with reusable blades decreased from 79.2% to 50.9%. Glottic exposure grades were mostly good, with CL grades of I or II independent of the pandemic. The overall FPS rate before and during the pandemic was 92.9% and 92.2%, respectively, with no significant change (*p* = 0.495). We observed no significant change in the FPS rate during the pandemic based on intubator grade. Approximately 100% of patients were successfully intubated within three attempts, and this rate was not affected by the pandemic (*p* = 0.133). In terms of intubation-related complications, cases of hypotension, hypoxemia, and cardiac arrest were slightly higher during the pandemic than before, but these differences were not statistically significant (*p* = 0.763, *p* = 0.187 and *p* = 0.228, respectively).

Figure 2 shows the changes in FPS rates according to intubator grade and type of airway difficulty. In the absence of airway difficulty, FPS did not change significantly during the pandemic, regardless of intubator grade (*p* = 0.713). For anatomically difficult airways, FPS was particularly low in the junior emergency physician (PGY1) group (78.9%) and showed a marginal decrease to 75.0% during the pandemic (*p* = 0.836). However, in the senior emergency physician (PGY ≥ 2) group, we found no change in FPS before and during the pandemic (*p* = 0.600). For physiologically difficult airway cases, FPS rates decreased during the pandemic period regardless of intubator grade, especially in the senior emergency physician group (98.0% to 88.5%; *p* = 0.006).

The results of univariate logistic regression analysis showed that pandemic protocol use, crash airway, anatomical difficulty, physiological difficulty, intubator grade, intubation method, device, and GEG were associated with the FPS rate. In the univariate analysis of FPS, none of the independent variables modified the effect of the homogeneity of the ORs. Table 3 summarizes the results of the multivariate logistic regression analysis used to predict the FPS rate of emergency ETI, considering the use of the COVID-19 pandemic intubation protocol. Application did not have a significant independent effect on the FPS rate (OR 0.72, *p* = 0.069). Among the patient-related variables, anatomical difficulty (OR 0.45, *p* < 0.001) and physiological difficulty were associated with a significantly lower FPS odds ratio (OR 0.52, *p* < 0.001). The FPS OR was 3.47 (*p* < 0.001) for an intubation grade ≥PGY 2. The method of intubation did not significantly affect the FPS; however, video laryngoscopy was more favorable for FPS than direct laryngoscopy, and mounted VL with a larger screen had the highest FPS OR (OR 2.16, *p* = 0.006). In addition, FPS was significantly lower with a glottic view of GEG III or IV (OR 0.52, *p* < 0.001).

## 4. Discussion

In this study, we sought to determine whether FPS and major complication rates in the emergency department changed following the implementation of a new infection-prevention intubation protocol. There have been a few reports on the outcomes of using new endotracheal intubation protocols during the pandemic period, but studies in more diverse ED settings are needed to conclude whether the use of infection prevention intubation protocols is associated with FPS of ETIs in the ED [10,12,25,26]. Some researchers have focused on intubator PPE donning and FPS [22]. However, the implementation of new protocols may also affect several factors related to the competence of the intubation team, including equipment, training, team size, and communication. As this was a retrospective observational study using registry data, it was not possible to collect data while controlling for each of these factors during the pandemic. However, this is a before-and-after comparison study in the same EDs, so the results of this study may provide some insight into the impact of implementing infection prevention intubation protocols on endotracheal intubation FPS in the ED.

The overall FPS rate of ETIs performed by emergency physicians during the pandemic remained at the level of FPS in the pre-pandemic period, which is similar to or slightly higher than findings from other studies [25,26,27]. In multivariate analysis, the infection prevention intubation protocol was not significantly associated with FPS. There are several possible reasons why there was no decrease in ETI FPS during the pandemic. First, PPE does not appear to have a significant effect on intubator performance. A number of studies in simulation-based and real-world clinical settings have shown that PPE for COVID-19 prevention does not reduce intubation success rates, even when it causes discomfort to intubators [26,27,33]. In addition, our ED staff had experienced several previous outbreaks in the ED, such as the Middle East respiratory syndrome, so they were familiar with the use of PPE. We also used a single PPE protocol, which may have helped ED staff adapt quickly to PPE. A study using separate PPE protocols for suspected and non-suspected COVID-19 patients reported relatively low FPS compared with our study [26]. Second, the pre-training we provided to successfully apply the new protocol may have been effective. Our primary concern with the COVID-19 protocols was human factors rather than the effect of PPE. Limited manpower and equipment are a challenge for intubation teams. In addition, the implementation of new protocols can place a cognitive burden on healthcare workers [34]. We conducted in-situ simulation training prior to protocol implementation to test the process and identify and correct factors that could lead to errors. Several studies have reported that in-situ simulation improves patient outcomes [25,35]. Thirdly, the high VL usage rate may have contributed to the sustained FPS levels. Multivariate analysis showed that VL was independently significantly associated with FPS, and several other studies have supported the relationship between high VL use and high FPS [15,20,25,27,30]. In addition, VLs have a clear advantage over DLs in the donning of PPE [25,36]. Although portable VLs with smaller screens were used significantly more during the pandemic period, the OR for FPS was not different from that for mounted VLs with larger screens (OR 2.12 vs. 2.16). Therefore, portable VLs may also be recommended for future pandemic protocols, such as mounted VLs.

Compared with the pre-pandemic period, there is a slight increase in intubation-related severe hypotension, hypoxemia, and cardiac arrest during the pandemic period. Studies have reported varying rates of intubation-related complications during the pandemic, making it difficult to conclude a relationship between the implementation of new protocols and intubation-related complications [25,26,37]. Although the implementation of new protocols may have influenced complications, this was more likely due to the disease rather than a change in protocol [26,38]. In addition, differences in the intubation teams, protocols, and equipment, as well as bias in the data in the medical records, may also contribute to the differences in the rates of intubation-related complications between the studies.

In a subgroup analysis, we examined the grade-specific FPS of intubators. During the pre-pandemic period, junior (PGY1) emergency physicians had a particularly low overall FPS compared with senior Emergency physicians, which is consistent with the findings of other studies [39,40,41]. In this study, the FPS rate for junior emergency physicians during the pandemic was 85.5%, which was not significantly lower than in the pre-pandemic period. There are several reasons for this. First, even during the pandemic, junior emergency physicians were the first attempt intubator in 22.3% of cases. A lack of experience in intubation may have contributed to the FPS rates of junior emergency physicians reaching pre-pandemic levels. Second, a focus on video laryngoscopy skills training and on-site airway training may have been effective [42,43,44]. Third, we had the junior emergency physicians discuss the intubation strategy with a senior emergency physician prior to intubation, which may have provided an additional educational benefit.

We also investigated the grade-specific FPS of intubators in anatomically and physiologically difficult airways. Multivariate analysis showed that anatomical/physiological difficulty was independently significantly associated with FPS, which is consistent with the results of other studies [45]. However, during the pandemic, the change in FPS rates for these two difficulties differed according to the grade of the intubator. For anatomically difficult airways, junior emergency physicians had particularly low FPS rates compared with senior emergency physicians, and this did not change significantly before or after the pandemic. Given the low FPS rate of junior emergency physicians, a senior emergency physician should be the first attempt intubator, at least in situations where an anatomically difficult airway is anticipated. For physiologically difficult airways, all emergency physicians had an acceptable FPS during the pandemic; however, senior emergency physicians had a significantly lower FPS (88.5%) than before the pandemic (98.0%). Possible reasons for this include: the study used hypoxia and hypotension as criteria for physiological distress, which may have been more severe in patients intubated by senior emergency physicians. Reduced team performance due to limited manpower is another possible cause [22,46,47]. Based on these findings, there is a need to supplement current intubation protocols to prevent infection in cases of anatomically and physiologically difficult airways.

In this study, we found several characteristic changes during the pandemic. First, fewer intubations were associated with injuries. This may be related to the overall decrease in the number of injuries during the pandemic [48,49,50,51,52,53]. Second, while anatomical difficulties did not change, physiological difficulties increased. The reasons for this are unknown, but one study raised the possibility that restrictions on access to emergency departments in South Korea during the pandemic may have delayed ED visits by critically ill patients [36].

The limitations of this study are First, this is a retrospective observational study, so we could not establish a causal relationship between the introduction of the new protocol and intubation outcomes. Second, we combined changes in several factors related to intubation into one variable, “introduction of new protocol”; therefore, we were not able to determine the individual impact of the different factors included in the change in protocol. By comparison with this study, this would be possible in another pandemic. Third, other factors not investigated in this study may have influenced the results. Fourth, the generalizability of the findings is limited because the study was conducted in the context of the experience of two emergency departments. The use of our strategy may also be limited in other ED settings. In addition, we did not examine the rate of intubation-related COVID-19 infection among staff involved in intubation. Other reports have shown that physicians who performed endotracheal intubation in patients with SARS-CoV-2 did not have a significant risk of COVID-19 infection, especially if they used appropriate PPE [26].

## 5. Conclusions

During the COVID-19 pandemic, the FPS rate and complications for adult ETI performed by emergency physicians using COVID-19 infection prevention intubation protocols were similar to those in the pre-pandemic period.

## Figures and Tables

**Figure 1 jpm-13-01017-f001:**
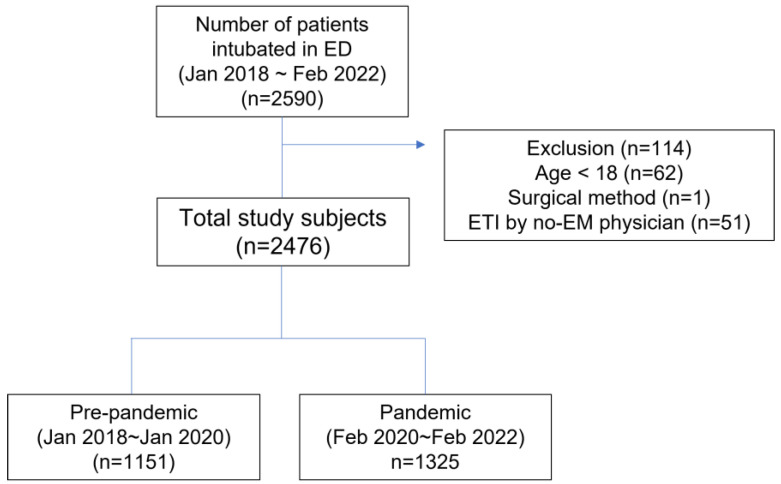
Study flowchart. ED, emergency department; EM, emergency medicine.

**Figure 2 jpm-13-01017-f002:**
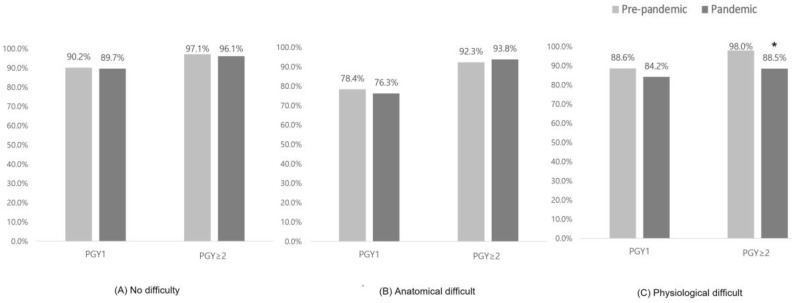
Comparison of first-pass success rates by intubator grade based on type of airway difficulty. * *p* < 0.05. PGY, postgraduate year.

**Table 1 jpm-13-01017-t001:** Baseline characteristics of participants.

Parameter	Pre-Pandemic (*n* = 1151)	Pandemic (*n* = 1325)	*p*-Value
Sex			
Male	730 (63.4%)	836 (63.1%)	0.866
Female	421 (36.6%)	489 (36.9%)	
Age, yr (median, IQR)	68 (55–79)	69 (56–78)	0.850
Indication			
Medical	922 (80.1%)	1125 (84.9%)	0.002
Injury	229 (19.9%)	200 (15.1%)	
Purpose			
Airway patency	739 (64.2%)	843 (63.6%)	0.885
Ventilation/oxygenation	214 (18.3%)	244 (18.4%)	
Prophylactic	198 (17.2%)	238 (18.0%)	
Crash airway	463 (40.2%)	545 (41.1%)	0.647
Predicted difficulty			
Anatomical	389 (33.8%)	432 (32.6%)	0.529
Physiologic	221 (19.2%)	299 (22.6%)	0.040
Emergency department			
A	752 (65.3%)	572 (43.2%)	<0.001
B	399 (34.7%)	753 (56.8%)	

**Table 2 jpm-13-01017-t002:** Differences in endotracheal-intubation-related variables between pre-pandemic and pandemic periods.

Variable	Pre-Pandemic (*n* = 1151)	Pandemic (*n* = 1325)	*p*-Value
Grade of intubators			
PGY 1	313 (27.2%)	296 (22.3%)	0.009
PGY 2	284 (24.7%)	320 (24.2%)	
PGY ≥ 3	554 (48.1%)	709 (53.5%)	
Method			
No medications	496 (43.1%)	572 (43.2%)	0.037
Awake technique	79 (6.9%)	60 (4.5%)	
RSI	576 (50.0%)	693 (52.3%)	
Device			
DL	111 (9.6%)	85 (6.4%)	<0.001
VL: portable/disposable	128 (11.1%)	566 (42.7%)	
VL: mounted/reusable	912 (79.2%)	674 (50.9%)	
GEG			
I –II	1093 (95.0%)	1271 (95.9%)	0.250
III–IV	58 (5.0%)	54 (4.1%)	
FPS	1069/11.51 (92.9%)	1221/1325 (92.2%)	0.495
PGY 1	272/272 (86.9%)	253/296 (85.5%)	0.347
PGY 2	267/284 (94.0%)	299/320 (93.4%)	0.432
PGY ≥ 3	530/554 (95.7%)	669/709 (94.4%)	0.178
Success in three attempts	1147 (99.7%)	1324 (99.9%)	0.133
Complication			
Hypotension *	91/688 (13.2%)	109/780 (13.9%)	0.763
Hypoxemia ^†^	135/688 (19.6%)	175/780 (22.4%)	0.187
Cardiac arrest ^‡^	1/688 (0.1%)	4/780 (0.5%)	0.228

PGY, postgraduate year; RSI, rapid sequence intubation; DL, direct laryngoscope; VL, video laryngoscope; GEG, glottic exposure grade; and FPS, first-pass success. * Excluding patients who had cardiac arrest or whose blood pressure was not measured after intubation.^†^ Only for patients who did not experience cardiac arrest and had a pre-intubation SpO_2_ > 90%. ^‡^ Excluding patients who had had cardiac arrest prior to intubation.

**Table 3 jpm-13-01017-t003:** Multivariate odds ratios for the factors associated with the first-attempt success rate for emergency intubation according to the COVID-19 pandemic intubation protocol.

	FPS(%)	OR	*p*-Value	CI
Pandemic				
Pre-pandemic	92.9	1		
Pandemic	92.2	0.72	0.069	0.51–1.01
Crash airway				
Non-crash	91.7	1		
Crash	93.7	1.19	0.886	0.52–2.12
Anatomical difficulty				
Not difficult	94.9	1		
Difficult	87.7	0.45	<0.001	0.35–0.68
Physiological difficulty				
Not difficult		1		
Difficult		0.52	<0.001	0.33–0.68
Grade of intubators				
* PGY1	86.2	1		
PGY ≥ 2	94.5	3.47	<0.001	2.47–4.71
Method				
No medication	93.4	1		
Awake	89.2	0.97	0.943	0.40–2.36
RSI	92.0	0.97	0.927	0.49–1.94
Device				
DL	81.6	1		
Portable VL	93.2	2.12	0.011	1.19–3.77
Mounted VL	93.5	2.16	0.006	1.24–3.45
GEG				
I–II	94.5	1		
III–IV	50.0	0.52	<0.001	0.36–0.76

* PGY, postgraduate year; RSI, rapid sequence intubation; DL, direct laryngoscopy; VL, video laryngoscopy; GEG, glottic exposure grade; FPS, first-pass success.

## Data Availability

The datasets analyzed during the study are available from the corresponding author upon reasonable request.

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
