# Peer review of "Introduction of Infection Prevention Tracheal Intubation Protocol during the COVID-19 Pandemic Is Not Associated with First-Pass Success Rates of Endotracheal Intubation in the Emergency Department: A Before-and-After Comparative Study"

_jpm, 2023, doi:10.3390/jpm13061017_

Round 1

Reviewer 1 Report

Dear authors

I received your submission as a reviewer and have read it several times with a great interest. It seems that you have a valuable data for presentation as a paper in a scientific journal. However, there are some minor points that I want to share with you. When I read the title, I focused on two words: “impact” and” novel”. Impact on what? Neither the title nor the introduction clearly answered this question. Indeed, this is a major ambiguity for those who start to read the paper. Therefore, I suggest you to revise the title appropriately, and also make the exact aim of the study at the end of introduction. Moreover, you pointed to “a novel protocol”. In my opinion, using the mentioned PPEs was a forced instruction from the authorities of your country. So, I believe that it is not good to call it as “novel”. In my opinion, it is better to use another title to show that you were seeking for what outcome on what population via which intervention. Considering these points, from my viewpoint, you need to revise the title and introduction. However, the rest of the paper and its content is appropriate.

Kind regards

Author Response

Dear reviewer 1

We completely agree and appreciate your comments. We were faced with two challenges during the COVID-19 pandemic: the application of unfamiliar intubation protocols that we had not previously experienced and reduced training opportunities for less-experienced emergency physicians. We tried to find a way to overcome these two major challenges and wanted to see the results, which is why we have revised the title of this paper as follows: “Comparison of endotracheal intubation outcomes in emergency departments before and after the introduction of infection prevention intubation protocols during the COVID-19 pandemic”And we’ve also significantly revised the introduction to reflect the new title. We also significantly revised the introduction.

Reviewer 2 Report

Dear authors,

The manuscript is scientific soundness. However, I have comments as follows:

Introduction:

- Please give more information on the knowledge gaps of publishing this manuscript. There have been a lot of articles regarding treatment modifications during the COVID-19, what are the interesting points that this paper will add to the society.

- I think the primary aim in the introduction part that stated 'we aimed to evaluate the impact of changes in airway management techniques before and during the pandemic on ETI behavior and performance in the ED.' did not match the primary outcome stated at method and material line 115-117.

The impact of changes in airway management techniques should be elaborated. Moreover, the change of management in ED would definitely impact the ETI behaviors and ED performance, how would this manuscript described things differently. Would impacts of change to patient outcomes be more interesting?

Method: It was hard to see the impact of change because the primary outcome that measured FPS also affected by other variables. Table 3 - multivariate analysis, please identify which variables were put into the model and why?

Additionally, as the study measured 2 centers, I wonder if it would be possible to calculate the causal inference of this retrospective study. For instances, find the instrument variables, or use difference in difference technique to try to infer the causation.

Discussion:Overall, the discussion part needed more references in a lot of reason given by the authors. I think there have been several studies reported the related issues in this manuscript. For example, Line 215-225, are there any references for those reasons the authors discussed.

The conclusion should be softened like the association between the change of management and FPS or patient outcomes since the studies was the cohort not the RCT. It would be inappropriate to conclude causation if the study was an observation.

English language had moderate quality. Minor revision is required.

Author Response

<Reviewer 2>

Dear reviewer 2

Thank you for taking the time to review our paper, and we respond to your comments below.

Introduction:

1) Please give more information on the knowledge gaps of publishing this manuscript. There have been a lot of articles regarding treatment modifications during the COVID-19, what are the interesting points that this paper will add to the society.

Emergency intubation is associated with an increased risk of adverse events, especially when multiple or prolonged attempts are made. Numerous algorithmic approaches have been published to mitigate these adverse events. The increased risk of transmission of COVID-19 to healthcare workers during intubation has led to a rapid shift from protocols focused solely on patient outcomes to those focused on the safety of both patients and healthcare workers. As a result, numerous new protocols and guidelines have been implemented for the safe intubation of patients with or at risk for COVID-19. Until now, these protocols have been site-specific, variable, and based on extrapolating previous knowledge to new clinical situations. As the pandemic has passed, several researchers have shared their experiences with intubation in the emergency department during the pandemic. We believe that there are still insufficient studies to draw consistent conclusions. A variety of site-specific protocols have been applied and their impact needs to be shared.

We have also tried to supplement the risks of intubation protocols that have been universally recommended during the pandemic to our site, and if successful, we believe that our results will help many EDs similar to ours to formulate intubation strategies that are effective and safe to achieve high FPS.

In this study, we found that the FPS and complications of ETIs performed during the COVID-19 infection prevention intubation protocol were not significantly different from those before the pandemic. This gives us confidence that we can perform relatively successful intubations in the emergency department despite the presence of several factors that we fear will adversely affect FPS. Based on this experience, we will be able to establish effective intubation protocols for the next infectious disease pandemic.

2) I think the primary aim in the introduction part that stated 'we aimed to evaluate the impact of changes in airway management techniques before and during the pandemic on ETI behavior and performance in the ED.' did not match the primary outcome stated at method and material line 115-117.

We agree with you. The wording we used was ambiguous. In the introduction, we have revised the primary objective to be clearer as follows.

“This study aimed to answer the question: In adult emergency department patients undergoing emergency intubation by emergency physicians, is the adoption of a site-specific COVID-19 infection prevention intubation protocol to mitigate the risk of intubation failure associated with first-pass success or other intubation-related outcomes?

This study aimed to answer the question: In adult emergency department patients undergoing emergency intubation by emergency medicine physicians, is the adoption of a site-specific COVID-19 infection prevention intubation protocol to mitigate the risk of intubation failure associated with first-pass success or other intubation-related outcomes?

3) The impact of changes in airway management techniques should be elaborated. Moreover, the change of management in ED would definitely impact the ETI behaviors and ED performance, how would this manuscript described things differently. Would impacts of change to patient outcomes be more interesting?

The intubation environment during the pandemic raises concerns that fewer team members, less training, and fewer opportunities will negatively impact FPS and outcomes. While we were unable to find literature on the optimal number of people needed for intubation, it is easy to assume that fewer staff than before the pandemic will be disadvantageous. For less experienced emergency physicians, FPS is traditionally lower, and without real-time supervision, we are concerned that the likelihood of failure will be even higher. Strict PPE and aerosol boxes can make it difficult for practitioners to operate equipment and communicate with staff. We were concerned about these negative effects of the protocol during the pandemic and took steps to address them, including pre-application training, focused training on essential skills such as video laryngoscope techniques, and pre-discussion. We were able to maintain high FPS levels under more unfavorable conditions than before the pandemic, and complications increased but were not significant. While it is difficult to prove causality between our efforts and the outcome, we can prioritize the application of methods such as our strategy in the event of a new pandemic.

4) Method: It was hard to see the impact of change because the primary outcome that measured FPS also affected by other variables. Table 3 - multivariate analysis, please identify which variables were put into the model and why?

We performed a stratified analysis among independent variables using univariate analysis and looked for the effect-modification factors based on the homogeneity of the odds ratio (OR). We identified all variables with P < 0.10 as candidate parameters for inclusion in the subsequent multivariate model. We used multivariate logistic regression to find variables that have an independent relationship with FPS. Odds ratios were reported with 95% confidence intervals (CIs) and P values. All reported CIs are 2-sided 95% intervals, and tests were performed at the 2-sided 5% significance level. On univariate logistic regression analysis, pandemic protocol application, crash airway, anatomic difficulty, physiologic difficulty, grade of intubators, intubation method, device, and GEG were associated with FPS. In the univariate analysis of FPS, none of the independent variables modified the effect of homogeneity of ORs.

5) Additionally, as the study measured 2 centers, I wonder if it would be possible to calculate the causal inference of this retrospective study. For instances, find the instrument variables, or use difference in difference technique to try to infer the causation.

Because this study used data from two centers, it is possible that differences between the centers may have intervened in the results. However, the two centers share residents, use the same equipment, have a joint airway management curriculum, and have used the same emergency airway management protocols for many years, so we believe there is less room for institutional differences to intervene in the results.

6) Discussion: Overall, the discussion part needed more references in a lot of reason given by the authors. I think there have been several studies reported the related issues in this manuscript. For example, Line 215-225, are there any references for those reasons the authors discussed.

In response to your comments, we have significantly revised the Discussion section by reviewing more references. We have also added literature on the epidemiology of trauma during the coronavirus pandemic to the reference list.

7) The conclusion should be softened like the association between the change of management and FPS or patient outcomes since the studies was the cohort not the RCT. It would be inappropriate to conclude causation if the study was an observation.

We agree with you. As this is an observational study, we cannot prove a causal relationship between the application of the protocol and ETI outcomes. We have revised our conclusions as follows:

"During the pandemic, FPS and complications of adult emergency ETIs performed by emergency medicine physicians using site-specific COVID-19 infection prevention intubation protocols were similar when compared to pre-pandemic."

8) Comments on the Quality of English Language

English language had moderate quality. Minor revision is required.

We have already had our English proofread by 'Editage', but as per your suggestion, we have improved the quality of our English once again by using MDPI's Language editing service.

Round 2

Reviewer 2 Report

Dear Authors,

Thank you for the extensive revision of the manuscript. The introduction was more convincing and the overall of the manuscript looks very much better. Statistical information were also well describe and the references added in many parts also made the manuscript more credible.

I have more comments as follows:

- English language after the revision was extremely hard to understand. Please revise the written languages.

- Please revise the title by adding primary outcome or most interesting finding to it.

- The study's aim was still not clearly identified or comprehended, which might be because of the language. 

- the revised version of discussion looked more confusing, please revise the first paragraph again by focusing on the primary outcome and aim of the study. Moreover, many parts in the discussion contradicted themselves, which make the study less credible.

There were too many confounding factors to conclude anything from these data. I suggested re-design the study and data collection with a more control way. For example, control the education to the intubation team and study the effect of the education, or control the intubation measures/process and study the success rates pre/post processes changes.

Required extensive revision.

Author Response

Dear reviewer,

We are very grateful for the time you have taken to review our manuscript.

And I have tried to improve the quality of the paper as much as possible based on your comments.

We've changed the English language to be as concise as possible.

We have changed the title to include the main findings.

We've rewritten the introduction to make the purpose of the study clearer.

Since Round 1, we've seen a lot of other research and thought about other factors that may affect FPS, so the discussion section has become a bit confusing. We've tried to make it as concise as possible.

During the pandemic, many variables related to intubation changed simultaneously; it was not possible to control for each of these individually, so we combined them into a single variable, "implementation of new protocols", for analysis. The issue of training (education) you mention is also an integral part of the protocol adoption process. So we believe this can be included in the ‘protocol adoption’ variable. Other observational studies using same research protocol as ours have also this limitation. Independent analysis of the individual factors included in the 'protocol' variable that changed during the pandemic will be possible through comparative studies with intubation data from the next pandemic.

Round 3

Reviewer 2 Report

Dear authors,

thank you for your revision.

The english language was much better, some small revision required